# Genetic Characterization and Pathogenesis of H5N1 High Pathogenicity Avian Influenza Virus Isolated in South Korea during 2021–2022

**DOI:** 10.3390/v15061403

**Published:** 2023-06-20

**Authors:** Ra Mi Cha, Yu-Na Lee, Min-Ji Park, Yoon-Gi Baek, Jae-In Shin, Chang Hwa Jung, Mingeun Sagong, Gyeong-Beom Heo, Yong-Myung Kang, Kwang-Nyeong Lee, Youn-Jeong Lee, Eun-Kyoung Lee

**Affiliations:** Avian Influenza Research & Diagnostic Division, Animal and Plant Quarantine Agency, 177 Hyeoksin 8-ro, Gyeongsangbuk-do, Gimcheon-si 39660, Republic of Korea; rami.cha01@korea.kr (R.M.C.);

**Keywords:** highly pathogenic avian influenza, H5N1, pathogenicity, genotype, chicken, ducks

## Abstract

High pathogenicity avian influenza (HPAI) viruses of clade 2.3.4.4 H5Nx have been circulating in poultry and wild birds worldwide since 2014. In South Korea, after the first clade 2.3.4.4b H5N1 HPAI viruses were isolated from wild birds in October 2021, additional HPAIV outbreaks occurred in poultry farms until April 2022. In this study, we genetically characterized clade 2.3.4.4b H5N1 HPAIV isolates in 2021–2022 and examined the pathogenicity and transmissibility of A/mandarin duck/Korea/WA585/2021 (H5N1) (WA585/21) in chickens and ducks. Clade 2.3.4.4b H5N1 HPAI viruses caused 47 outbreaks in poultry farms and were also detected in multiple wild birds. Phylogenetic analysis of HA and NA genes indicated that Korean H5N1 HPAI isolates were closely related to Eurasian viruses isolated in 2021–2022. Four distinct genotypes of H5N1 HPAI viruses were identified in poultry, and the majority were also found in wild birds. WA585/21 inoculated chickens showed virulent pathogenicity with high mortality and transmission. Meanwhile, ducks infected with the virus showed no mortality but exhibited high rates of transmission and longer viral shedding than chickens, suggesting that they may play an important role as silent carriers. In conclusion, consideration of both genetic and pathogenic traits of H5N1 HPAI viruses is required for effective viral control.

## 1. Introduction

The high pathogenicity avian influenza (HPAI) H5 virus of the A/Goose/Guangdong/1/1996 (Gs/Gd) lineage has been circulating in birds for more than 25 years, causing huge economic impacts globally in regions including Europe, Africa, Asia, and North America [1,2,3,4,5]. Due to consistent circulation in poultry and wild birds, several different clades and subtypes of the Gs/Gd H5 virus have emerged [3]. Since 2014, H5 viruses belonging to clade 2.3.4.4, including H5N6, H5N8, and H5N1, have spread globally, causing frequent outbreaks in poultry and threatening both industry and public health [6]. In particular, clade 2.3.4.4b H5Nx HPAI viruses have been consistently isolated in European and Asian countries since 2016 [6,7]. In late 2021, clade 2.3.4.4b H5N1 HPAI viruses were first detected in wild birds in North America, and outbreaks in poultry farms have subsequently been reported [4,5].

In South Korea, H5Nx clade 2.3.4.4 HPAI viruses have been detected in wild birds and caused outbreaks in poultry farms since 2014. Subsequently, different subtypes or subclades of clade 2.3.4.4 H5 viruses were newly introduced with divergent genetic features. These viruses display varying levels of pathogenicity and transmissibility in experimental infections of chickens and ducks [8,9,10,11,12]. Clade 2.3.4.4b H5N8 HPAI viruses isolated in 2020–2021 in South Korea were genetically similar to European viruses identified in 2020, which were classified into two major genotypes (G1 and G2) [12]. The 2020/2021 isolate of the Korean H5N8 HPAI virus was highly virulent and transmissible in specific pathogen-free (SPF) chickens. Meanwhile, virus-inoculated ducks showed no mortality with longer viral shedding and higher rates of transmission [13].

The new reassortants of clade 2.3.4.4b H5N1 HPAI viruses have emerged after the epidemic caused by H5N8 HPAI in Europe during late 2020 and have become the pre-dominant HPAI circulating in poultry and wild birds [6,7,14]. In South Korea, the first novel clade 2.3.4.4b H5N1 HPAI virus was isolated from mandarin duck in October 2021 and, subsequently, from quail farms in November 2021. These H5N1 HPAI isolates were closely related to European H5N8 and H5N1 viruses but have a different internal gene configuration compared to the European isolates [15]. After the first case report, more H5N1 HPAI outbreaks occurred in poultry farms, and wild bird cases have, subsequently, been detected in South Korea. In this study, we genetically characterized clade 2.3.4.4.b H5N1 HPAI virus isolates from South Korea in the 2021–2022 winter season and investigated the pathogenicity and transmission of the index H5N1 HPAI isolate (A/mandarin duck/Korea/WA585/2021(H5N1)) in SPF chickens and ducks.

## 2. Materials and Methods

### 2.1. Viruses

A total of 53 H5N1 HPAI virus isolates were genetically characterized in this study. During the 2021–2022 winter season, H5N1 HPAI viruses from 47 outbreaks in poultry farms and 6 wild bird cases were isolated by the Animal and Plant Quarantine Agency (APQA), South Korea (Appendix A). For wild birds, a DNA barcoding system utilizing mitochondrial DNA from feces was used to determine the host species as previously described [16]. A/mandarin duck/Korea/WA585/2021(H5N1) (WA585/21) was the first H5N1 HPAI virus isolated from a wild bird in October 2021, and it was used to test for pathogenicity and transmissibility in SPF chickens and ducks [15].

### 2.2. Sequencing, Phylogenetic Analysis and Genome Constellation

Viral RNA was extracted from all HPAI virus isolates using the Patho Gene-spin DNA/RNA Extraction Kit (iNtRON Biotechnology, Seongnam South Korea) before being amplified by RT-PCR [10]. PCR products were purified with AMPure XP magnetic beads (Beckman Coulter, Brea, CA, USA). Multiplexed, paired end sequencing libraries were generated using the Nextera™ DNA Flex Library Prep Kit (Illumina, San Diego, CA, USA) and following the manufacturer’s instructions. Complete genome sequencing was performed using the Illumina MiSeq next generation sequencing platform (Illumina, San Diego, CA, USA). Genome sequences were directly assembled using the CLC Genomics Workbench software (Qiagen) and Bioedit. Sequences of the viruses isolated in this study were deposited in the GISAID database (http://platform.gisaid.org (accessed on 8 May 2023)), and the accession number of each virus is listed in Appendix A. Hemagglutinin (HA) and neuraminidase (NA) gene sequences from the 53 H5N1 HPAI isolates were subject to phylogenetic analysis. A reference dataset for constructing phylogenetic trees based on HA and NA sequences was obtained from the EpiFlu database from the Global Initiative on Sharing All Influenza Data (GISAID, https://www.gisaid.org). This dataset included HA gene sequences from H5NX viruses detected in Eurasia and North America in 2020–2022 and NA gene sequences from H5N1 viruses detected in Eurasia in 2017–2022, respectively (Appendix A). Maximum likelihood (ML) phylogenies were generated with 1000 bootstrap iterations using MEGA 6.0. A monophyletic cluster was defined by high bootstrap support (> 70%). For genome constellation, the genetic diversity of H5N1 HPAIVs was determined using RAxML version 8.2.12 with a gamma distribution and a general time reversible model [17]. All AIV sequences, collected from 1 January 2012 to 5 January 2022 (accessed on 12 May 2022), were obtained from the GISAID and GenBank databases and used as the dataset for the phylogenetic analysis of each internal gene. Genotypes were analyzed according to tree topology, and a nucleotide sequence identity of > 97% was considered significant when the bootstrap support value was > 70.

### 2.3. Animal Experiment

A total of 46 five-week-old SPF chickens (*Gallus gallus domesticus*) were obtained from CAVAC (Daejeon, Republic of Korea), and 36 two-week-old ducks were obtained from a commercial duck farm. All birds were negative for avian influenza (AI) antibodies and were housed in negative-pressure, high-efficiency, air-filtered isolation cabinets within a biosafety level 3 facility. Water and feed were provided ad libitum. An intravenous pathogenicity index (IVPI) test with SPF chickens was performed according to the World Organization of Animal Health (WOAH) guidelines to confirm virus pathogenicity [18]. SPF chickens were divided into four groups (*n* = 5) and intranasally inoculated with 0.1 mL of serial 10-fold dilutions ranging from 10^3^ to 10^6^ mean egg infectious dose (EID_50_) of WA585/21(H5N1) to determine the mean chicken lethal dose (LD_50_). For ducks, the mean bird infectious dose (BID_50_) was examined with three groups (n = 5) of intranasally inoculated ducks administered 10^2^, 10^4^, and 10^6^ EID_50_/0.1 mL of the virus. To determine viral transmissibility in chickens and ducks, 3 naïve chickens or ducks were cohoused with the 10^6^ EID_50_ virus inoculated group 8 h later. An additional 3 birds were inoculated intranasally with 10^6^ EID_50_/0.1 mL of the virus to examine viral replication in internal organs at 3 days post inoculation (dpi). Serum samples were collected at 14 dpi, and a hemagglutination inhibition (HI) test was performed. EID_50_ and BID_50_ were calculated according to the method of Reed and Muench [19]. All animal experiments were reviewed and approved by the Institutional Animal Care and Use Committee of the APQA (approval no: 2020-561).

### 2.4. Viral Shedding, Virus Replication in Organs, and Hemagglutination Inhibition (HI) Test

Oropharyngeal (OP) and cloacal (CL) swabs were collected at 1, 2, 3, 4, 5, 6, 7, 10, and 14 dpi in the 10^6^ EID_50_ inoculated and contact groups to evaluate viral shedding. Three birds from the inoculated group were euthanized and necropsied, twelve organs (trachea, thymus, heart, lung, kidney, brain, pancreas, cecal tonsil, liver, spleen, muscle, and proventriculus) were collected, and virus replication was assessed in each organ. For virus isolation, each OP and CL swab sample was suspended in 1 mL of Dulbecco’s Modified Eagle Medium (Gibco; Invitrogen, Carlsbad, CA, USA) containing antibiotics (Invitrogen), and each tissue sample was homogenized (*w*/*v* ratio of 10%). Samples were then centrifuged at 3500 rpm for 5 min, and the supernatant was titrated in chicken embryo fibroblasts (DF-1) to determine the 50% tissue culture infective dose (TCID_50_). The viral titer was calculated using the method of Reed and Muench [19]. The limit of viral detection was < 1 log10 TCID_50_/0.1 mL. Statistical significance was determined using Student’s *t*-test for independent samples with Graphpad Prism 5 (San Diego, CA, USA). The HI test was performed using standard procedures according to the WOAH instructions, using serum samples from the remaining live birds at 14 dpi, as described previously [13].

## 3. Results and Discussion

Since the first detection of clade 2.3.4.4b H5N1 HPAI in October 2021, viruses have continuously been detected in wild birds and have caused outbreaks in poultry farms. From October 2021 to March 2022, a total of 66 cases were detected in wild birds by the APQA (*n* = 6) and the National Institute of Wildlife Disease Control and Prevention (NIWDC) (*n* = 60) [20]. In poultry, a total of 47 H5N1 HPAI viruses have been isolated from chicken (*n* = 20), duck (*n* = 23), minor poultry (*n* = 4; Korean native chicken (*n* = 2), and quail (*n* = 2)) farms from November 2021 to April 2022. Most H5N1 HPAI outbreaks in poultry and wild birds were found in the western regions of the country, where there are many migratory bird habitats and intense poultry farming areas (based on the Korean Animal Health Integrated System (KAHIS)) (Appendix A). The regional occurrence of HPAI in 2021–2022 agreed with previous findings in Korea [21]. In this study, we examined the genetic characteristics of 53 HPAI viruses isolated from poultry (*n* = 47) and wild birds (*n* = 6) (Appendix A). All H5N1 isolates possessed multibasic amino acid sequences (PLRERRKR*GLF) at their HA cleavage site, indicating high pathogenicity avian influenza (Appendix A).

Phylogenetic analysis of the HA gene revealed that H5N1 HPAI viruses were closely related to recent HPAI viruses isolated from neighboring Asian countries, including Japan and China, and other Eurasian viruses isolated in 2020–2021. Korean H5N1 HPAI isolates formed a distinct phylogenetic group from North American HPAI viruses isolated in 2021 (Figure 1A). HA genes from H5N1 HPAI viruses isolated in the 2021–2022 winter season were also classified as a G2 cluster and are closely related to the clade 2.3.4.4b H5N8 HPAI viruses described previously [12]. For the NA gene, Korean H5N1 viruses clustered into a genetic subgroup with European and Asian viruses isolated in 2021 (Figure 1B). Our phylogenetic analysis results were correlated with those of wild bird H5N1 HPAI viruses isolated in Japan in 2021–2022, showing that Korean H5N1 HPAI viruses seemed to be closely related to the HPAI viruses isolated in southern Japan [22]. Interestingly, it seems like the major HA gene subgroup of H5N1 HPAI viruses of Korea and Japan were different, which may be due to their geological location and the migration routes of the affected wild bird. [22]. These genetic differences between HPAI isolated from two neighboring countries in East Asia may suggest an influx of multiple genetically diverse HPAIV by migrating bird populations during the 2021–2022 winter season [22,23].

We further characterized the gene constellation of all H5N1 HPAI viruses through phylogenetic analysis of all eight gene segments (Figure 1 and Appendix A). H5N1 HPAI viruses isolated in 2021–2022 were classified into four distinct genotypes (Table 1). All genotypes comprised similar HA and NA genes, which were closely related to those present in 2020–2021 Eurasian HPAI viruses (Figure 2). Further, the genotypes contained reassorted internal genes, which may originate from Eurasian LPAI viruses. Genotypes I–III had the same internal gene constellations, with the exception of the PB1, PA, and NP genes. In genotype IV isolates, only PA genes clustered with those from genotype I–III viruses (Figure 2 and Table 1). Moreover, HA and NA genes from genotype IV viruses were closely related to H5N1 HPAI viruses isolated in China and other Asian countries [23,24]. In poultry, genotype I was mainly detected for autumn to early winter (October–December 2021) and was followed by genotype II, which became the major genotype from January till early April 2022 (Figure 2 and Table 1). In addition, genotypes III and IV were sporadically detected between December 2021 and March 2022 (Figure 2 and Table 1). For wild birds, genotype I and II HPAI viruses were identified from six wild birds by the APQA, and genotype IV, in addition to two novel genotypes, were identified in wild birds following examination by the NIWDC (unpublished data). Our gene constellation analysis demonstrated that multiple genotypes were introduced into the country, and one genotype become a major virus causing disease outbreaks for a certain period of time, with the sporadic introduction of other genotype viruses, which were similar to those isolated in European countries in the 2021–2022 winter season [25]. In addition, European H5N1 HPAI viruses isolated in 2021 showed that multiple genotypes were detected within two distinct HA gene sub-lineages of clade 2.3.4.4b [7]. We speculate that these H5N1 HPAI viruses from Europe and other Eurasian viruses underwent genomic rearrangements in wild bird breeding sites or an unknown location. Additionally, these novel reassortant viruses were probably introduced into Korea by migrating wild bird populations, which was confirmed by the detection of the same genotype in both wild birds and poultry outbreaks. Further genetic analysis is needed to understand the role of wild birds in the evolution of HPAI viruses with genetic reassortment.

The pathogenicity and transmissibility of WA585/21(H5N1), the first Korean H5N1 HPAI isolate, were evaluated in SPF chickens and ducks. The IVPI value in chickens was 2.98, classifying the virus as HPAI by the WOAH standard (Table 2) [18], which is consistent with HA gene cleavage site analysis results. WA585/21(H5N1) inoculated chickens (10^6^ EID_50_) showed clinical signs of depression, greenish diarrhea, and neurological signs at 1–3 dpi. Furthermore, we observed 100% mortality in inoculated birds, with an MDT of 2.6 days and an LD_50_ of 10^3^.^7^ EID_50_. In the contact group, the rate of transmission was 100%, and all three birds died following the exposure (Table 2). No HI response was observed in the surviving SPF chickens. Viral shedding in chickens via the OP and CL routes was observed at 1–4 dpi, with a peak viral titer of 10^4.8^ and 10^5.8^ TCID_50_/0.1 mL at 4 dpi in the OP and CL, respectively (Figure 3). In contact birds, viral shedding by the OP and CL routes was detected until 6 dpi and peak titers were comparable with those of the inoculated group (Figure 3). At 3 dpi, WA585/21 exposed chickens displayed viral replication in all internal organs, with viral titers of 10^3.1–4.9^ TCID_50_/0.1 mL (Figure 4). WA585/21(H5N1) had lower MDT and LD50 in chickens than clade 2.3.4.4.b H242(H5N8) isolated in the 2020–2021 winter season in Korea had. Furthermore, WA585/21 inoculated chickens showed longer periods of viral shedding and higher rates of onward transmission in contact birds than H242(H5N8) showed [13]. These results suggest that WA585/21 is more virulent in chickens with a higher transmission rate than the previously isolated H242/2020(H5N8), and these phenotypes may affect viral dissemination in the field. A recent study demonstrated that Japanese H5N1 HPAI isolates from the 2021–2022 winter season had slightly different pathogenicity and transmissibility in chicken infections than our study, which may be due to different experimental design and use of commercial chickens [26].

In ducks, virus-inoculated birds (10^6^ EID_50_) showed no mortality, but minor clinical signs of greenish diarrhea were observed at 3–5 dpi. The mean bird infectious dose (BID_50_) was 10^3^.^2^ EID_50_ (Table 2). All ducks, except the 10^2^ EID_50_ inoculated group, were seroconverted following infection. Virus-inoculated ducks displayed viral shedding via the OP and CL route, which lasted for 5 days (1–5 dpi) with a peak viral titer at 2–3 dpi (10^3.2–4.4^). The contact group also showed viral shedding for 3 to 4 days (Figure 3). Viral replication was detected in the trachea (10^3.9^) and lung (10^2.6^), whereas other internal organs contained low viral titers (10^1.2–2.2^). Viral titers in the thymus, liver, heart, lung, brain, and pancreas were significantly lower than in chickens (*p* < 0.05) (Figure 4). Our data suggest that in ducks, WA585/21 is less virulent but has high rates of transmission and longer periods of viral shedding than SPF chickens. These observations in ducks are consistent with previous findings with Korean clade 2.3.4.4b viruses, confirming that ducks may play an important role as a silent carrier in the field in terms of HPAI transmission [13]. A recent phylodynamic analysis of clade 2.3.4.4 H5N8 HPAI viruses isolated in Korea in 2014–2016 also demonstrated that domestic ducks played a key role in viral transmission and maintenance after multiple introductions from wild birds [21]. Further, several studies have demonstrated that ducks were likely to transmit HPAI H5 viruses from wild birds to poultry sectors in the fields [27,28,29].

In this study, we investigated the genetic and pathogenic features of H5N1 HPAI viruses isolated in Korea during the 2021–2022 winter season. Phylogenetic analysis revealed that clade 2.3.4.4b H5N1 HPAI viruses were closely related to Eurasian viruses isolated in 2020–2021 and consisted of four distinct genotypes, which were distributed across both poultry and wild birds, indicating multiple introductions to Korea from diverse wild bird populations. WA585/21 were highly virulent in inoculated chickens, which displayed high levels of mortality and onward transmission. In ducks, WA585/21 infection caused no mortality but had higher rates of transmission and viral shedding. In conclusion, an understanding of both genetic and pathogenic features of H5N1 HPAI viruses will be important to establish efficient biosecurity measures and surveillance programs to prevent viral spread in the field.

## Figures and Tables

**Figure 1 viruses-15-01403-f001:**
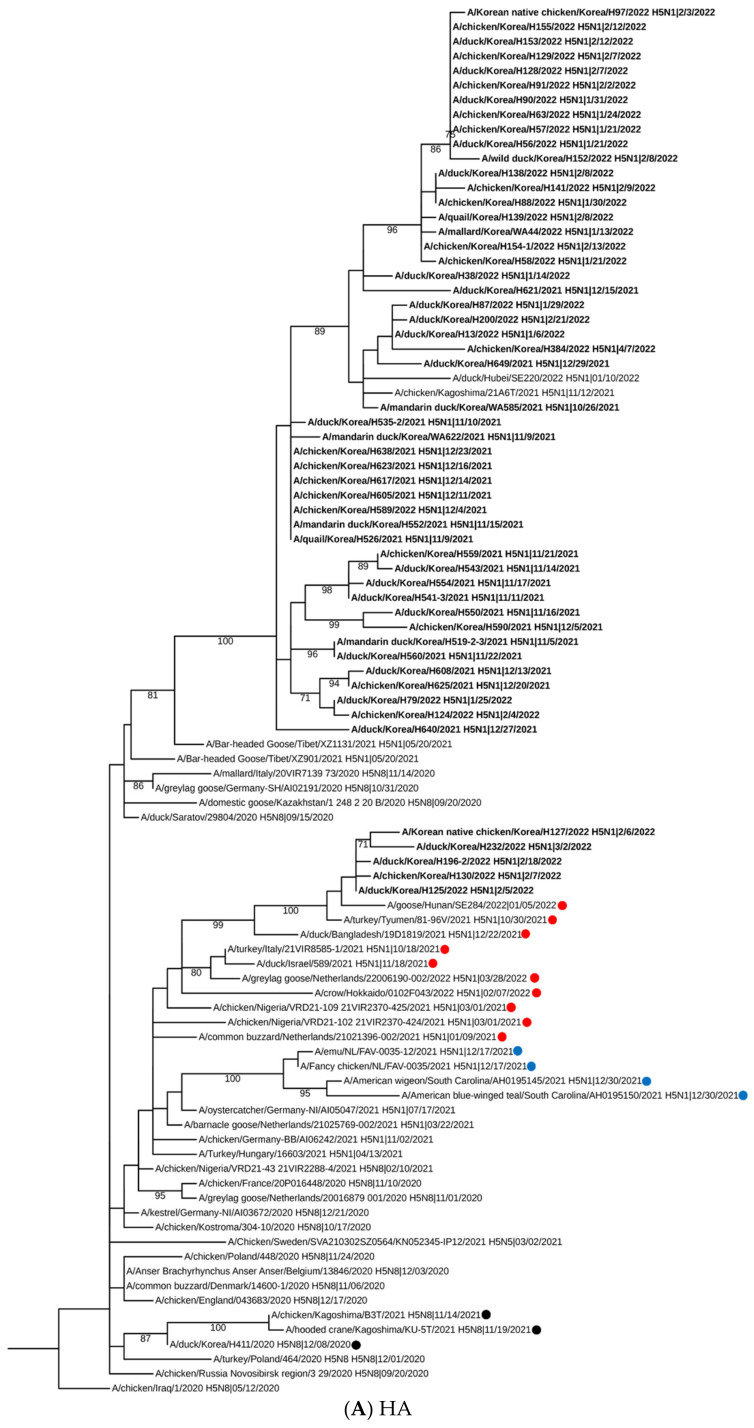
Maximum likelihood (ML) phylogenetic tree of hemagglutinin and neuraminidase genes from Korean H5N1 HPAI viruses isolates in the 2021–2022 winter season. ML phylogenetic trees based on the hemagglutinin (**A**) and neuraminidase (**B**) gene sequences were constructed using MEGA 6.0. The scale bar represents the number of nucleotide substitutions per site. A monophyletic cluster was defined when bootstrap values (1000 replicates) were > 70%. H5N1 HPAIVs isolated in Korea in 2021–2022 are highlighted in bold. Red dot: Eurasian HPAI strains isolated in 2021–2022; blue dot: North American HPAI strains isolated in 2021; black dot: H5N8 HPAIVs isolated in Korea /Japan in 2020–2021.

**Figure 2 viruses-15-01403-f002:**
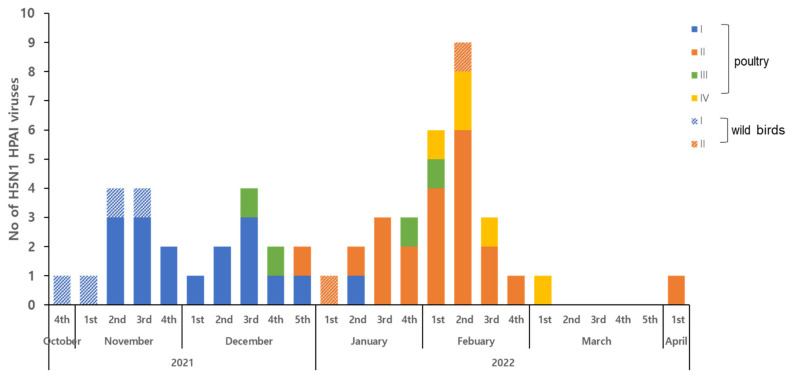
Gene constellation analysis of Korean H5N1 HPAI viruses isolated from wild birds (*n* = 6) and poultry (*n* = 47) in 2021–2022. Detailed gene constellation analysis of H5N1 HPAI viruses was performed using phylogenetic analysis of all eight gene segments. Genotypes of H5N1 HPAI virus isolates are shown by isolation date.

**Figure 3 viruses-15-01403-f003:**
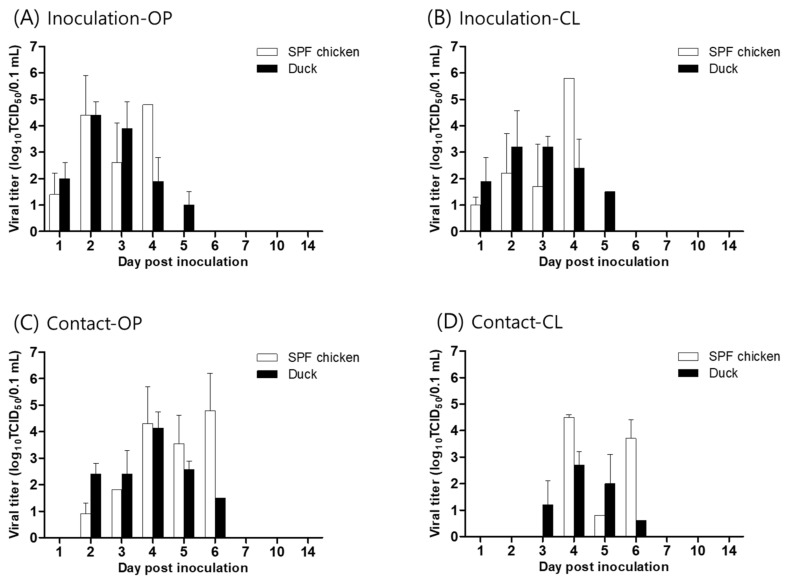
Virus isolation from oropharyngeal (OP) (**A**,**C**) and cloacal (CL) (**B**,**D**) swab samples of WA585/21 exposed SPF chickens and ducks. Birds were intranasally inoculated with 10^6^ EID_50_/0.1ml of the virus (**A**,**B**). Three naïve birds (n = 3) were cohoused with the inoculated group 8 h after primary infection (**C**,**D**). Viral titers are shown as the mean ± standard deviation.

**Figure 4 viruses-15-01403-f004:**
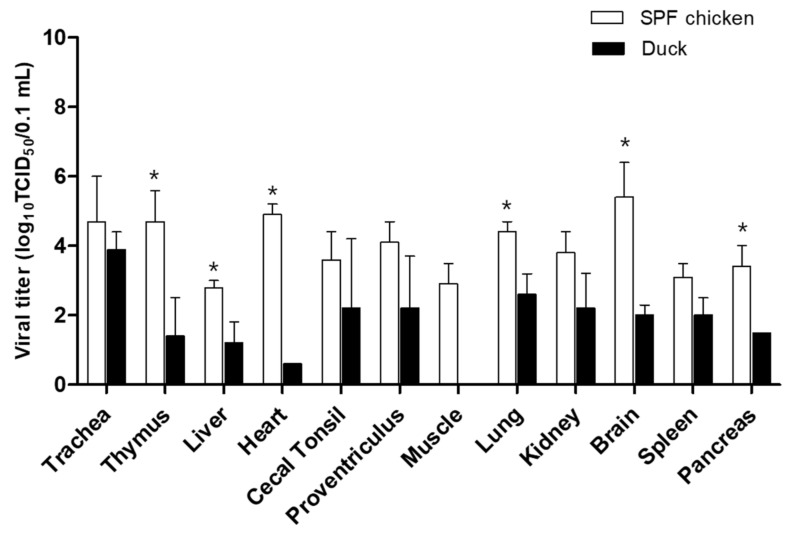
Virus titer in various organs from chickens and ducks at 3 dpi. Birds were intranasally inoculated with 10^6^ EID_50_/0.1 mL of the virus, internal organs from three birds were collected at 3 dpi, and viral titers were determined in DF-1 cells. The data shown indicate the average of calculable positive viral titer from each organ. ***: *p* < 0.05.

**Table 1 viruses-15-01403-t001:** Genome constellation of H5N1 (clade 2.3.4.4b) HPAI in South Korea in 2021–2022.

Virus Name	Subtype	Reassortant	Genotype	Phylogenetic Group within Each Gene Segment							
				PB2	PB1	PA	HA	NP	NA	MP	NS
21WA585	H5N1	AAA5A1AA	I	A	A	A	5	A	1	A	A
21WA622	H5N1	AAA5A1AA	I	A	A	A	5	A	1	A	A
22WA44	H5N1	ADA5A1AA	II	A	D	A	5	A	1	A	A
21H519-2-3	H5N1	AAA5A1AA	I	A	A	A	5	A	1	A	A
21H526	H5N1	AAA5A1AA	I	A	A	A	5	A	1	A	A
21H535-2	H5N1	AAA5A1AA	I	A	A	A	5	A	1	A	A
21H541-3	H5N1	AAA5A1AA	I	A	A	A	5	A	1	A	A
21H543	H5N1	AAA5A1AA	I	A	A	A	5	A	1	A	A
21H550	H5N1	AAA5A1AA	I	A	A	A	5	A	1	A	A
21H552	H5N1	AAA5A1AA	I	A	A	A	5	A	1	A	A
21H554	H5N1	AAA5A1AA	I	A	A	A	5	A	1	A	A
21H559	H5N1	AAA5A1AA	I	A	A	A	5	A	1	A	A
21H560	H5N1	AAA5A1AA	I	A	A	A	5	A	1	A	A
21H589	H5N1	AAA5A1AA	I	A	A	A	5	A	1	A	A
21H590	H5N1	AAA5A1AA	I	A	A	A	5	A	1	A	A
21H605	H5N1	AAA5A1AA	I	A	A	A	5	A	1	A	A
21H608	H5N1	ACB5C1AA	III	A	C	B	5	C	1	A	A
21H617	H5N1	AAA5A1AA	I	A	A	A	5	A	1	A	A
21H621	H5N1	AAA5A1AA	I	A	A	A	5	A	1	A	A
21H623	H5N1	AAA5A1AA	I	A	A	A	5	A	1	A	A
21H625	H5N1	ACB5C1AA	III	A	C	B	5	C	1	A	A
21H638	H5N1	AAA5A1AA	I	A	A	A	5	A	1	A	A
21H640	H5N1	AAA5A1AA	I	A	A	A	5	A	1	A	A
21H649	H5N1	ADA5A1AA	II	A	D	A	5	A	1	A	A
22H013	H5N1	ADA5A1AA	II	A	D	A	5	A	1	A	A
22H038	H5N1	AAA5A1AA	I	A	A	A	5	A	1	A	A
22H056	H5N1	ADA5A1AA	II	A	D	A	5	A	1	A	A
22H057	H5N1	ADA5A1AA	II	A	D	A	5	A	1	A	A
22H058	H5N1	ADA5A1AA	II	A	D	A	5	A	1	A	A
22H063	H5N1	ADA5A1AA	II	A	D	A	5	A	1	A	A
22H079	H5N1	ACB5C1AA	III	A	C	B	5	C	1	A	A
22H087	H5N1	ADA5A1AA	II	A	D	A	5	A	1	A	A
22H088	H5N1	ADA5A1AA	II	A	D	A	5	A	1	A	A
22H090	H5N1	ADA5A1AA	II	A	D	A	5	A	1	A	A
22H091	H5N1	ADA5A1AA	II	A	D	A	5	A	1	A	A
22H097	H5N1	ADA5A1AA	II	A	D	A	5	A	1	A	A
22H124	H5N1	ACB5C1AA	III	A	C	B	5	C	1	A	A
22H125	H5N1	BBA5B1BB	IV	B	B	A	5	B	1	B	B
22H127	H5N1	BBA5B1BB	IV	B	B	A	5	B	1	B	B
22H128	H5N1	ADA5A1AA	II	A	D	A	5	A	1	A	A
22H129	H5N1	ADA5A1AA	II	A	D	A	5	A	1	A	A
22H130	H5N1	BBA5B1BB	IV	B	B	A	5	B	1	B	B
22H138	H5N1	ADA5A1AA	II	A	D	A	5	A	1	A	A
22H139	H5N1	ADA5A1AA	II	A	D	A	5	A	1	A	A
22H141	H5N1	ADA5A1AA	II	A	D	A	5	A	1	A	A
22H152	H5N1	ADA5A1AA	II	A	D	A	5	A	1	A	A
22H153	H5N1	ADA5A1AA	II	A	D	A	5	A	1	A	A
22H154-1	H5N1	ADA5A1AA	II	A	D	A	5	A	1	A	A
22H155	H5N1	ADA5A1AA	II	A	D	A	5	A	1	A	A
22H196-2	H5N1	BBA5B1BB	IV	B	B	A	5	B	1	B	B
22H200	H5N1	ADA5A1AA	II	A	D	A	5	A	1	A	A
22H232	H5N1	BBA5B1BB	IV	B	B	A	5	B	1	B	B
22H384	H5N1	ADA5A1AA	II	A	D	A	5	A	1	A	A

**Table 2 viruses-15-01403-t002:** Pathogenicity and transmissibility of WA585 in SPF chickens and ducks.

Bird Species	IVPI	Virus Dose (EID_50_/0.1 mL)	Mortality (%)	MDT (Day)	HI Titer (log2, Mean ± SD)	LD_50_ (EID_50_/0.1 mL)	BID_50_ (EID_50_/0.1 mL)
SPF chicken	2.98	10^6.2 a^	5/5 ^b^ (100)	2.6	NT	10^3.7^	-
		10^5.2^	5/5 (100)	4.2	NT		
		10^4.2^	5/5 (100)	6.0	NT		
		10^3.2^	0/5 (0.0)	-	0/5(0.0)		
		10^2.2^	0/5 (0.0)	-	0/5(0.0)		
		Contact	3/3 (100)	4.6	NT		
Duck	-	10^6.2^	0/5 (0.0)	-	4/4 *(6.5 ± 1.3)	-	10^3.2^
		10^4.2^	0/5 (0.0)	-	5/5 (5.8 ± 1.1)		
		10^2.2^	0/5 (0.0)	-	0/5 (0.0)		
		Contact	0/3 (0.0)	-	3/3 (5.4 ± 1.5)		

^a^: Birds were intranasally inoculated with serial 10-fold dilutions, ranging from 10^3.0^ to 10^6.0^ EID_50_/0.1 mL. The virus inoculum was confirmed with back titration, and the exact dose was shown. ^b^: The number of affected birds/birds per group. IVPI, intravenous pathogenicity index; LD50, mean lethal dose; MDT, mean death time; NT, not tested; BID_50_, mean bird infectious dose. * One duck dead for an unknown reason at 13 dpi (AIV test negative).

## Data Availability

The data presented in this study are available in the Global Initiative on Sharing All Influenza Data (GISAID). All accession numbers of the virus sequence data were listed in Appendix A.

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
