# Peer review of "Genetic Characterization and Pathogenesis of H5N1 High Pathogenicity Avian Influenza Virus Isolated in South Korea during 2021–2022"

_viruses, 2023, doi:10.3390/v15061403_

Round 1
Reviewer 1 Report
This paper reports the genetic and pathogenicity analyses of high pathogenicity avian influenza viruses of the H5N1 subtype isolated in Korea during the winter of 2021-2022. Genetic analysis of 53 H5N1 virus isolates was performed for eight gene segments, and the pathogenicity of the isolated virus to chickens and ducks was further examined. This is a comprehensive analysis of the viruses that circulated in East Asia last season, which is valuable information for future epidemics in this area and the world. The supplemental data are also very transparent and adequately present useful information to the reader. I hope that the following minor points will be improved to enhance the quality of this paper.
1. Figure 1
In the analysis of HA and NA genes, the respective subgroups should be clearly indicated. In particular, the subgroups of viruses that were prevalent in Europe and North America during the 2021–2022 season should be clearly indicated. The subgroups of viruses isolated in Korea during the 2020–2021 season should also be clearly indicated (e.g., the position of G1 and G2 groups).
2. The authors should carefully review the paper before resubmitting it. There are many careless mistakes in the text, as shown below.
L37 two spaces
L42 ex-per
L56 subse-quently
L99 scientific name in italics
L112 10 to the sixth power, superscript
L179 A1?
L231 6 dpi
L305 Author list abbreviations are incorrect.
3. Titles of cited references
There are several papers where the first letter of every word is shown in capital letters.
This should be corrected.
Basic Format
Journal Articles:
1. Author 1, A.B.; Author 2, C.D. Title of the article. Abbreviated Journal Name Year, Volume, page range.
Reviewer 2 Report
Avian influenza viruses of high pathogenicity (PAI) of the 2.3.4.4 H5Nx clade have been circulating among domestic and wild birds worldwide since 2014. In South Korea, after the first HPAI virus of the 2.3.4.4 H5N1 clade was isolated from wild birds in October 2021, additional HPAIV outbreaks occurred in poultry farms until April 2022. In this study, the authors genetically characterized the H5N1 HPAI isolates of the 2.3.4.4b clade in 2020-2021 and studied the pathogenicity and the possibility of transmission of the A/mandarin duck virus/Korea/WA585/2021(H5N1) (WA585/21) in chickens and ducks. The proposed scientific article is of interest to virologists and veterinarians. The data obtained by the authors and presented in this article are important for understanding the evolution of a highly pathogenic influenza virus and the development of antiepizootic measures. I believe that the article can be accepted into the journal without revision.
